# WS-iFSD: Weakly Supervised Incremental Few-shot Object Detection Without Forgetting

Xinyu Gong[1], Li Yin[3], Juan-Manuel Perez-Rua[2], Zhangyang Wang[1], Zhicheng Yan[2]

[1]University of Texas at Austin, [2]Meta, [3]Sylph AI

Traditional object detection algorithms rely on extensive annotations from a pre-defined set of base categories, leaving them ill-equipped to identify objects from novel classes. We address this limitation by introducing a novel framework for Incremental Few-Shot Object Detection (iFSD). Leveraging a meta-learning approach, our hypernetworkis designed to generate class-specific codes, enabling object recognition from both base and novel categories. To enhance the hypernetwork's generalization performance, we propose a Weakly Supervised Class Augmentation technique that significantly amplifies the training data by merely requiring image-level labels for object localization. Additionally, we stabilize detection performance on base categories by freezing the backbone and detection heads during meta-training. Our model demonstrates significant performance gains on two major benchmarks. Specifically, it outperforms the state-of-the-art ONCE approach on the MS COCO dataset by margins of $2.8\%$ and $20.5\%$ in box AP for novel and base categories, respectively. When trained on MS COCO and cross-evaluated on PASCAL VOC, our model achieves a four-fold improvement in box AP compared to ONCE.

## 1. Introduction

Despite great advances in object detection in the past years [2–13], a large portion of those approaches only tackle many-shot detection problem, and use abundant but expensive manual annotations to train object detectors in a batch manner. The resultant model is only able to detect objects from a closed set of base categories. Recently, there is a surge of few-shot object detection approaches, which can detect objects from novel categories using only few-shot examples [14–25]. In particular, we are interested in a more challenging setting called *Incremental Few-Shot object Detection (iFSD)* [26], where the detector is required to incrementally recognize more novel categories in an online manner while still being able to detect objects from base categories. iFSD approaches can be divided into two categories, depending on whether model fine-tuning is required to recognize novel categories.

Fine-tuning-based approaches require fine-tuning the model using the training samples of both base- and novel categories, and often suffer from the catastrophic forgetting of base categories [16, 20–22]. The demanding requirements of accessing the base category data and running model back-propagation also make them unsuitable for on-device deployment with limited storage and computation power. In contrast, fine-tuning-free methods [14, 23, 26] waive such requirements, and our approach falls into this category. We adopt a recent meta-learning framework ONCE [26], which meta-trains *a hypernetwork* to generate class codes for novel categories on the fly without model back-propagation. However, due to the limited number of base object categories and adversely changed detection head during meta-training, ONCE struggles with both *poor generalization to novel categories* and *inferior performance on base categories*.

To this end, we propose a novel weakly supervised approach *WS-iFSD*, which greatly improves the meta-training of hyper-network by augmenting it with many weakly localized objects from one order of magnitude more object categories. An off-the-shelf object localization model Grad-CAM [1] is adopted to coarsely localize the objects of interest using only image-level labels in the ImageNet [27]. This increases the number of classes by $18\times$ and the number of images by $6\times$. Due to the largely augmented classes and images, we demonstrate even with such coarsely inferred bounding

First Conference on Parsimony and Learning (CPAL 2024).

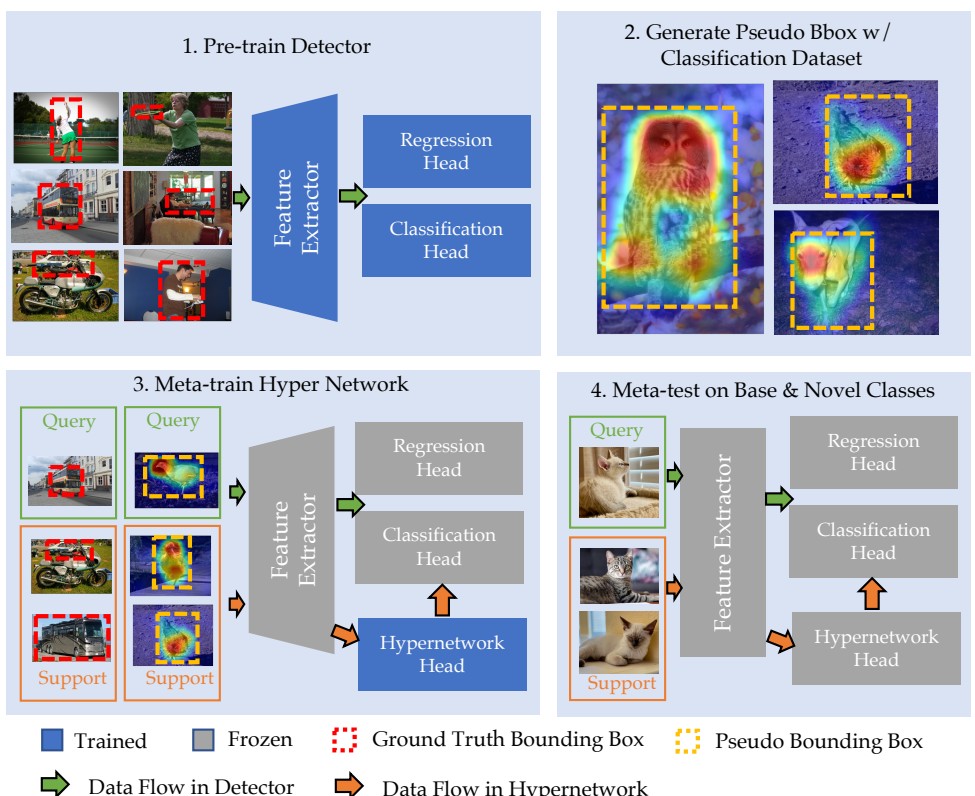

Figure 1: **The pipeline of WS-iFSD approach.** First, the entire detector is pre-trained on many-shot annotations of base categories. Second, we generate a large number of pseudo object annotations in the classification dataset using Grad-CAM [1]. Third, we combine human-labeled boxes of base categories and weakly-labeled boxes of augmented categories to mete-train the hypernetwork. Finally, the support images of novel categories are fed into the hypernetwork, which generates the classification weights for the detector to recognize the novel categories.

boxes the hyper-network is meta-learned to generate significantly better class codes for novel categories. WS-iFSD also addresses the issue of inferior detection performance for base categories by freezing the backbone and detection head during meta-training. By doing this, after meta-training we are able to *completely* preserve the superior detection performance on base categories achieved in the stage of pre-training on many-shot annotations of base categories. An overview of the proposed WS-iFSD approach is shown in Figure 1.

We extensively evaluate WS-iFSD on two object detection benchmarks, including MS COCO [28] and PASCAL VOC [29]. On MS COCO dataset, we improve the state-of-the-art approach ONCE [26] by remarkably large margins of 2.8% and 20.5% box AP for novel and base categories, respectively. On PASCAL VOC dataset, without any fine-tuning, our WS-iFSD model outperforms ONCE [26] by over $4\times$ more mAP.

To summarize, this work makes the following contributions.

- We tackle the challenging *Incremental Few-Shot Detection* problem with the constraint that model back-propagation is not allowed when few-shot examples of novel categories are received. A novel class augmentation approach is proposed to increase the number of object categories by $18\times$, and the number of images by $6\times$ during the meta-training of hypernetwork. It substantially improves its generalization performance to the novel categories.

- We build a simple yet strong baseline model where the backbone is shared by the object detector and the hypernetwork. It is accompanied with an improved pre-training recipe to

improve the detection of base categories. We freeze the detector backbone during meta-training and reuse the class codes of base categories from the pre-training to *fully* preserve the high detection performance for base categories.

- We extensively evaluate the proposed WS-iFSD approach on 2 benchmarks including MS COCO, and PASCAL VOC. WS-iFSD consistently outperforms other state-of-the-art approaches.

## 2. Related work

**Object detection**. Deep learning models for object detection include both two-stage [2, 30–32] and one-stage models [7, 11, 13, 33–37]. The training of both types of detectors is often data-inefficient, demanding the availability of large-scale human-annotated datasets [28, 29, 38, 39]. Also those detectors only recognize object categories seen in the training data, and extending them to handle novel categories is difficult or simply impossible after deployment. This work tackles the Incremental Few-Shot object Detection (iFSD) problem, where the model is required to recognize novel categories with only few-shot examples after model deployment.

**Few-shot image classification.** Few-shot learning (FSL) methods aim to solve the data-inefficiency problem and produce reliable classifiers from a few training samples. Many methods have been proposed for image classification task [40–50]. However, many findings from image classification FSL do not transfer to the FSD problem. For example, an effective method for FSL of image classification is to learn a strong feature extractor [51, 52] by simply pushing up the classification accuracy on a base dataset of labeled images. This strategy has found to be counter-productive for FSD, where simply maximizing detection performance on a base dataset leads to inferior transferability to novel categories [26]. This work focuses on the iFSD problem, and aim to improve the detection performance for both novel- and base categories.

**Few-Shot object Detection (FSD).** Existing FSD methods often require to access the data of both base- and novel categories for iteratively finetuning the model to recognize novel categories [15, 16, 53–55]. This incurs a large storage and computational cost that is prohibitive for on-device deployment. Also, fine-tuning based approaches often suffer from so-called *catastrophic forgetting*, where the detection performance on base categories degrades significantly [15, 16, 54, 56, 57]. A one-time novel category enrollment is possible but cannot be easily extended to incrementally enroll subsequent novel categories [16, 58]. The recent ONCE method [26] tackles the iFSD problem. However, it only uses the data of limited base categories for meta-learning, and achieves inferior performance for both novel- and base categories. We exploit large-scale image classification dataset with only image-level labels, and employ the off-the-shelf Grad-CAM technique to coarsely localize the object boxes for augmenting the iFSD meta-training with more object categories.

**Weakly Supervised Object Localization (WSOL).** A wide variety of CNN models are shown to be able to generate class-discriminate activation regions after being only trained on a classification dataset [59–63]. For example, Grad-CAM [1] uses the gradients of the target image-level category at the final convolutional layer to coarsely localize the region of the target object. It allows us to collect a tremendously large number of coarse object boxes from image classification dataset without human annotation. Our work adopts Grad-CAM to infer object boxes for up to 1000 object categories in the ImageNet dataset, and use them to augment the classes during the model meta-training.

**Weak-Supervision for Object Detection (WSOD).** Conventional WSOD approaches only use image-level annotations to train object detectors [64–69]. Many of them leverage image-level labels to build soft annotations through multiple instance learning (MIL) [70–72]. More recent works exploit WSOL approaches, such as Grad-CAM, to improve the object box proposal [73], and detector pre-training [74]. However, to our best knowledge, WSOL has not been adopted for the iFSD problem. In this work, we demonstrate that by augmenting the iFSD model meta-training with weakly localized abundant object boxes from a large number of object categories, we are able to boost the detection performance on novel categories.

# 3. Approach

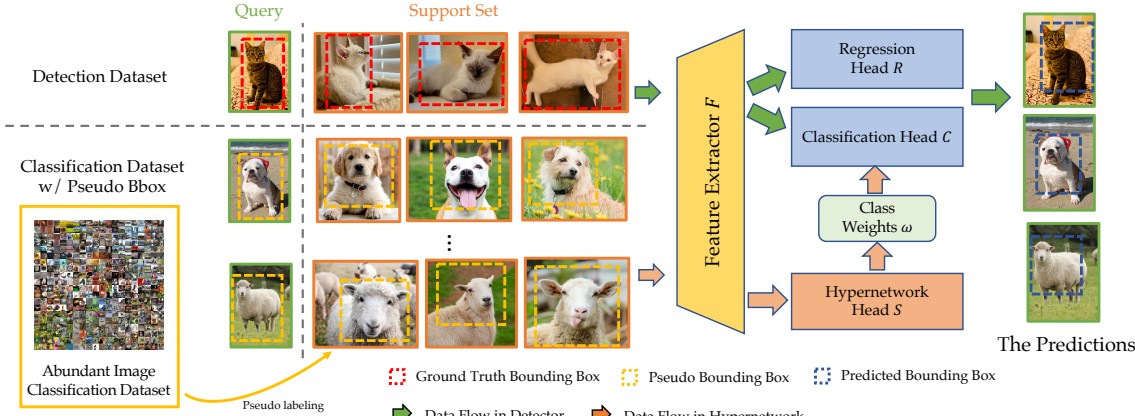

Figure 2: **The meta-training and meta-testing of WS-iFSD.** The detector network is based on FCOS [11]. The centerness branch is omitted for simplicity. The backbone and both heads are pre-trained on the many-shot dataset. The hypernetwork is composed of feature backbone and hypernetwork head. **Meta-training**: The hypernetwork head $S$ is meta-trained while other network parts are frozen. **Meta-testing**: The support set of a novel class is fed into the hypernetwork to produce the class weight $\omega$. The output weight is then imprinted into the existing classification head $C$, enabling the detector to recognize novel class samples (query).

## 3.1. Incremental Few-Shot Object Detection Task

For completeness, we first briefly review the iFSD task introduced in ONCE work [26], which requires the system to recognize two types of object categories. First, it needs to detect objects from base categories, which have abundant annotations in the training stage. Second, it can be updated on the fly to recognize novel object categories unseen in the training stage given only few-shot examples. To support efficient on-device deployment where the storage and computational resources are limited, we further refrain the model from using computationally expensive back-propagation to recognize novel categories.

## 3.2. Motivations

To support the learning of novel categories, ONCE does not require iterative model fine-tuning. It only employs a hyper-network to take few-shot examples as input, and uses efficient model forward passes to generate the class codes for novel categories. Despite being the pioneering approach to handle the iFSD problem, it achieves inferior performance on both novel- and base categories, which can be attributed to the two drawbacks below.

**Constraints of Limited Object Categories in Meta-Training:** The model achieves a meager $1\%$ box AP on 20 novel COCO categories with 5-shot examples, as indicated in Table 2. We hypothesize that this suboptimal performance primarily stems from the limited variety of base categories (e.g., 60 COCO categories) employed during the meta-training of the hypernetwork. Such a constraint leads to overfitting on the base categories and subsequently results in poor generalization to novel classes. To address this limitation, we introduce a novel weakly-supervised class data augmentation technique. This approach leverages an off-the-shelf Grad-CAM model to infer object bounding boxes in a heuristic manner, thereby extending the diversity of object categories during meta-training. Specifically, this method capitalizes on a substantially broader set of categories available in image classification datasets, reaching up to 1,000 categories in ImageNet.

**Large performance drop on base categories**. ONCE adopts CenterNet detector [13], but only achieves $17.9\%$ base category box AP (Table 2 in [26]), which is more than $10\%$ lower than the original CenterNet results. This is caused by the early stopping during the pre-training of the de-

tector on the abundant annotations of base categories, which is claimed to be effective to prevent the feature extractor overfitting base categories (see footnote[5] in [26]), while the dedicated design of the over-parameterized hyper network intensifies the overfit on base categories. However, we hypothesize that it is unnecessary to use early stopping to trade the detection performance on base categories for that on novel categories, and it is possible to *fully* preserve the superior detection performance on base categories achieved during model pre-training. We design a simple Base-iFSD model containing a hyper network with restricted hypothesis space to validate this.

### 3.3. An Overview of WS-iFSD Pipeline

The pipeline of the proposed WS-iFSD is shown in Figure 1. In the first step, we pre-train the entire detector on many-shot annotations of base categories. We choose to use one-stage FCOS [11] detector with ResNet-50 backbone due to its simplicity and high efficiency. It uses separate localization- and classification heads to localize objects and predict categories, respectively. Second, we generate pseudo object bounding boxes for a much larger set of object categories in the ImageNet dataset. Third, we combine the weakly generated pseudo bounding boxes and human-annotated bounding boxes of base categories to meta-train the hypernetwork with other model modules frozen. Last, our iFSD model can conduct meta-testing, where the class codes of novel categories are generated by the hypernetworkand added to the classification head. The model is able to recognize both base- and novel categories.

### 3.4. Weakly-Supervised Class Data Augmentation

The meta-training of the hypernetwork is conducted in an episodic manner [75]. In each episode, we randomly sample an iFSD task that uses a class label set of $t$ labels. When the total number of base category labels is limited (e.g. only 60 base categories in MS COCO dataset), the hypernetwork is less likely to generalize well to novel categories. Therefore, unlike traditional data augmentation approaches, which only augment the training samples of existing categories by large-scale jittering [76], copy-paste transformation [77] or generative models [78], we propose to augment the meta-training data with more new categories, which would also greatly increase the number of iFSD tasks. However, collecting a large number of images and annotating bounding boxes of the objects from many more categories is prohibitively expensive. On the other side, image-level object labels are cheaper to obtain, and existing image classification datasets (e.g. ImageNet) often have a much larger scale than the object detection datasets (e.g. MS COCO, PASCAL VOC) in terms of the number of both images and categories.

**Pseudo-box generation with only image-level labels**. To leverage abundant image-level labels in the image classification datasets, we propose to use Grad-CAM to annotate the bounding box for them. Grad-CAM [1] is a visualization technique producing a coarse localization map highlighting the focusing region of a classification model, which is achieved by leveraging gradients of a target category flowing into the final convolutional layer. Concretely, given a classification dataset $D_{\mathrm{cls}}$, the labeling of object bounding boxes can be decomposed into two steps. (*1*) Train a normal classification model on the classification dataset $D_{\mathrm{cls}}$. (*2*) Use the trained model to generate the heat map of images in $D_{\mathrm{cls}}$. The value of generated heatmap ranges from 0 to 1. We select a highlighted area from the heatmap whose value is above a threshold $\beta_T$. Then a pseudo bounding box is generated based on the highlighted area, which is the minimal rectangle containing it. Examples of weakly localized objects are shown in Figure 3.

**Mis-Classification Filtering**. We empirically find Grad-CAM has difficulty in localizing the objects when the original image classification model makes an incorrect image-level prediction (See Figure 3). To reduce the noise in our pseudo-box generation with only image-level labels, we introduce a simple filtering technique, namely *mis-classification filtering* (**MCF**). We compare the object label predicted by the classification model with the ground-truth label, and exclude those misclassified images from our class data augmentation.

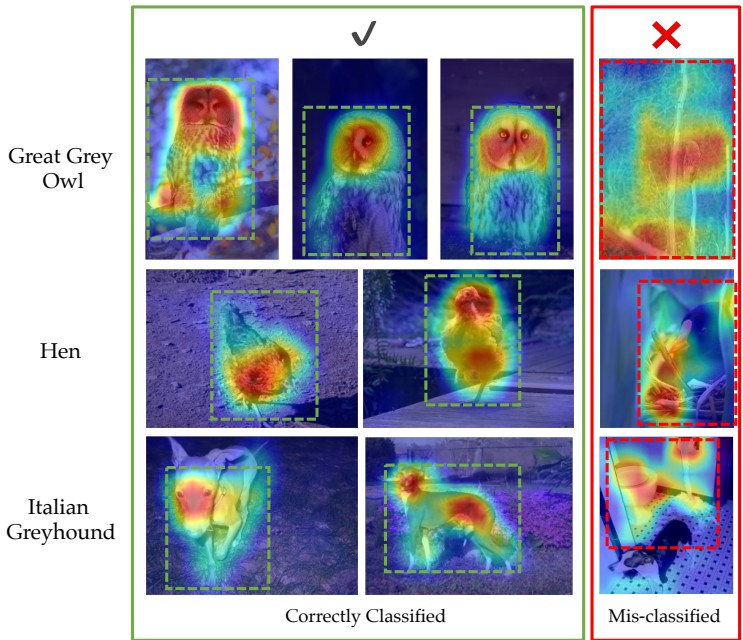

Figure 3: **The visualization of weakly localized boxes by GradCam [1]**. The right column denotes images removed by the proposed Mis-Classification Filtering, where the Grad-CAM model makes incorrect box predictions.

## 3.5. Base-iFSD without Catastrophic Forgetting

To achieve high detection performance on the base categories, we also propose a baseline iFSD model, namely Base-iFSD, which is designed to tackle the incremental few-shot detection problems *without forgetting base categories*. The Base-iFSD model consists of two parts: a FCOS [11] detector and a class weight hypernetwork.

### 3.5.1. Background on FCOS

FCOS is a simple anchor-free one-stage object detection model, achieving competitive results on several benchmarks. It is mainly composed of two parts: a feature extractor $F$ and 3 detection heads (see Figure 2). The feature extractor $F$ consists of a ResNet-50 [79] backbone and a feature pyramid network [80], which is used to extract the feature of input images $X$. There are 3 detection heads, including a box regression head $R$, a centerness head $T$, and a classification head $C$. The regression head $R$ is used to predict the distances from the location to four sides of the bounding box. The centerness head $T$ predicts a centerness score for the current pixel w.r.t to the object box, which can be used to suppress these low-quality detected bounding boxes. The classification head $C$ is used to predict the category label of the object box at the current pixel, and is parameterized by the classification weight $\alpha_C \in \mathbb{R}^{n \times m \times k \times k}$. Here $n, m$ denote the input and output channel width respectively, and $k$ the kernel size. For more details of the FCOS detector, please refer to the original work [11].

### 3.5.2. Hypothesis-Space-Restricted Hypernetwork

The class weight hypernetwork $H$ aims to generate the corresponding class-specific classification weight $\omega_c$ for the class of the input support set images, where $\omega_c \in \mathbb{R}^{n \times 1 \times k \times k}$. $c$ is the class of the input support set images. The newly generated classification weight $\omega_c$ for novel categories could be combined with the existing weight $\alpha_C$ for base categories in the classification head $C$ later.

Different from previous work ONCE [26], which uses a separate ResNet-50 backbone in the class weight hypernetwork, we adopted a lightweight design where the hypernetwork shares the backbone with the feature extractor $F$, and only uses a dedicated hypernetwork head $S$, which is only a

single convolutional layer. During the training of the hypernetwork, only the hypernetwork head $S$ is optimized, while the feature extractor $F$ is frozen. This design significantly reduces the issue of over-fitting by restricting the hypothesis space of hypernetwork through weight-sharing [81]. Besides, it also helps reduce the memory footprint tremendously. Formally, the output class weight $\omega_c$ of the hyper network could be written as:

$$\omega_c = \frac{1}{q} \sum_{j=1}^{q} H(X_j) = \frac{1}{q} \sum_{j=1}^{q} S(F(X_j)), \tag{1}$$

where $q$ is the number of images of the input support set. The output class weight can be imprinted into the classification head $C$, and enables the detector to recognize the unseen novel class $c$.

## 3.6. WS-iFSD with Strong Generalizability

We combine the aforementioned weakly-supervised class augmentation and Base-iFSD model to obtain the final WS-iFSD model. The training of both Base-iFSD and WS-iFSD model can be divided into two stages, including a pre-training stage for optimizing the feature extractor and detection heads, and a meta-training stage that optimizes the class weight hypernetwork. We elaborate on the pre-training, the meta-training and the meta-testing below.

**Pre-training**. At this stage, a regular FCOS model is trained on a sample-abundant object detection dataset, using the samples that only come from the base categories. The feature extractor $F$ and detection heads (*i.e.* $R$, $T$, and $C$) are optimized using the original training recipe in FCOS. More pre-training details are in the Appendix A.3.1.

**Meta-training with class augmentation**. At this stage, the parameters of the feature extractor and detection heads are frozen, and only the hypernetworkis optimized. This stage aims to train the hypernetwork to synthesize the corresponding classification weight for the class of the input support set. Following the approach in [26], we adopt an episodic meta-training strategy [75]. A set of base object categories are randomly sampled form an episodic task to mimic the meta-testing time where the support set with few-shot examples of novel categories is received. For Base-iFSD, only human annotations of objects from base categories are used to meta-train the hypernetwork. In contrast, WS-iFSD combines the pseudo-labeled augmented data with human annotations of objects from base categories to meta-train the hypernetwork, which could significantly enhance its generalizability.

In our meta-training scenario, we consider a distribution over tasks $p(\mathcal{T})$ on a mix of human-labeled base categories and pseudo-labeled augmented categories, and require the class weight hypernetwork to be able to generate the class weight for each of them. Concretely, the meta-training consists of two loops. In the inner loop, a task $\mathcal{T}_i$ is sampled from $p(\mathcal{T})$. The hypernetwork $H$ predicts the class weight with $K$ support set samples as input, then tested on new samples from $\mathcal{T}_i$ and feedback at the outer loop. The hypernetwork is then updated by considering the detection loss on the new samples from the sampled task. Note that the weakly-labeled object annotations is mixed with the human-labeled annotations during meta-training. More details can be found in the supplement.

**Meta-testing**. During meta-testing, each test query image comes with $K$ support set images ($K$-shot). The support set images are fed into hypernetwork $H$ generating the corresponding classification weight, which can be imprinted into the classification head to recognize novel categories. After that, the test query image goes through the detector (feature extractor and detection heads), and the detection prediction can be made.

# 4. Implementation details

## 4.1. Benchmarks

We conduct our experiments on two popular benchmarks, PASCAL VOC [29] and COCO [28], which have 20 and 80 classes, respectively. For each dataset, we split its classes into two main cate-

| Class-Aug | # Classes | # Images | # Objects |
|:---:|:---:|:---:|:---:|
| × | 60 | 98K | 364K |
| ✓ | 1,060 | 598K | 864K |

Table 1: **Comparing meta-training datasets after introducing weakly-supervised class augmentation**. COCO is used as the base object detection dataset. 1K classes from ImageNet with 500 images per class are used to construct the augmentation dataset.

gories, namely base classes, and novel classes. For COCO dataset, 20 classes are split as novel classes while the other 60 classes are treated as base classes [26]. For PASCAL VOC, we adopt the same setting with previous work [26]. We conduct a cross dataset experiment on it where we also use the 60 COCO base categories as the base classes for PASCAL VOC, and the whole 20 classes of PASCAL VOC are used as the novel classes, since they do not overlap.

## 4.2. Pseudo-labeled Dataset

To create a pseudo-labeled object detection dataset, we leverage Grad-CAM to annotate bounding boxes within the context of a large-scale classification dataset, specifically ImageNet [27]. ImageNet consists of 1,000 classes, each containing a substantial number of samples, making it an ideal choice for our purposes. We employ a ResNet-50 model pre-trained on ImageNet [79] as the base architecture for Grad-CAM. To generate activation heatmaps, we set an empirical threshold value $\beta_T = 0.6$ for highlighting relevant regions within an image. We construct our augmented dataset using all classes from ImageNet-1K, limiting to 500 images per class. A summary of the dataset statistics can be found in Table 1. Upon introducing the augmented data, we observe that, without any manual annotation, the number of classes, images, and objects in our dataset increased by factors of $17.6\times$, $6\times$, and $2.4\times$, respectively.

As COCO serves as our primary benchmark dataset, we meticulously examine the overlap of object categories between COCO and ImageNet. Our investigation reveals that 47 classes in ImageNet coincide with 20 of COCO's novel classes. These intersecting classes are designated as *novel overlapping classes*, while those without any overlap are referred to as *novel non-overlapping classes*. The potential impact of this overlap on performance metrics will be analyzed in Appendix A.2.3.

## 5. Experimental Results

For the main results we obtained in this section, we use all classes from ImageNet to construct the augmented data, where 500 images are randomly picked for each class after applying MCF. Comprehensive ablation studies have been conducted to substantiate the efficacy of each component within WS-iFSD. For a detailed discussion of these experiments, please refer to Appendix A.2.

## 5.1. Results on COCO Dataset

The comparison with existing state-of-the-art methods on the COCO dataset is summarized in Table 2. Remarkably, our Base-iFSD model achieves performance comparable to that of existing state-of-the-art methods on novel classes even without utilizing class data augmentation. It also surpasses the ONCE [26] model on base classes with a significant improvement of **20.5% AP**. This remarkable performance can be attributed to our hypernetwork design that freezes the backbone and recycles weights from a pre-trained feature extractor. This strategic constraint effectively reduces the hypothesis space, thus mitigating the risk of overfitting on base classes. Consequently, there is no need to employ early stopping during the training of the feature extractor to preserve its generalizability, a practice suggested in [26] that could compromise performance on base classes. After integrating our proposed weakly-supervised class data augmentation technique, we observed a nearly **4-fold** improvement in performance under 5-shot and 10-shot settings for novel classes. Notably, our 1-

| Shot | Method | Novel Classes mAP | Base Classes mAP |
|---|---|---|---|
| 1 | MAML [82] | 0.1 | N/A |
| | Feature-Reweight [14] | 0.1 | 2.5 |
| | ONCE [26] | 0.7 | 17.9 |
| | Base-iFSD (ours) | 0.8 (+0.1) | **38.4 (+20.5)** |
| | WS-iFSD (ours) | **2.3 (+1.6)** | **38.4 (+20.5)** |
| 5 | MAML [82] | 0.4 | N/A |
| | Feature-Reweight [14] | 0.8 | 3.3 |
| | ONCE [26] | 1.0 | 17.9 |
| | Base-iFSD (ours) | 1.0 | **38.4 (+20.5)** |
| | WS-iFSD (ours) | **3.8 (+2.8)** | **38.4 (+20.5)** |
| 10 | MAML [82] | 0.8 | N/A |
| | Feature-Reweight [14] | 1.5 | 3.7 |
| | ONCE [26] | 1.2 | 17.9 |
| | Base-iFSD (ours) | 1.2 | **38.4 (+20.5)** |
| | WS-iFSD (ours) | **4.0 (+2.8)** | **38.4 (+20.5)** |

Table 2: **iFSD results on COCO val2017 dataset.** We report detection metric box AP under 1, 5, and 10 shot test settings. The delta value is calculated based on previous SOTA work ONCE.

| Method | 1-Shot nAP | 5-Shot nAP | 10-Shot nAP |
|---|---|---|---|
| MAML [82] | - | 0.6 | 1.0 |
| ONCE [26] | - | 2.4 | 2.6 |
| Base-iFSD (ours) | 3.1 | 4.7 (+2.3) | 5.1 (+2.5) |
| WS-iFSD (ours) | **6.3** | **9.8 (+7.4)** | **11.2 (+8.6)** |

Table 3: **Cross dataset evaluation results on PASCAL VOC 2007.** We report the box mAP for novel categories under 1, 5, and 10 shot test settings. Our WS-iFSD outperforms other methods by a large margin. The delta value is calculated based on previous SOTA work ONCE. Here "nAP" is short for "novel classes mAP".

shot performance with WS-iFSD surpasses the previous state-of-the-art 10-shot results by a factor of **2**.

## 5.2. Transferability on PASCAL VOC Dataset

To demonstrate the generalizability of our model, we further conducted a cross-dataset experiment from COCO to PASCAL VOC. We test the performance of our method on novel categories of PASCAL VOC using Base-iFSD and WS-iFSD models we obtained in the last section directly, without any further adaptation. The results are shown in Table 3. The proposed WS-iFSD outperforms existing work by a large margin under all shot settings. Specifically, WS-iFSD surpasses the previous SOTA work ONCE by $4.1\times$, $4.3\times$ under 5 and 10 shot settings, respectively. Even the 1-shot result has surpassed the 10-shot result of ONCE, confirming our method's strong generalizability.

## 6. Conclusions

This paper introduces WS-iFSD, a novel framework that addresses the pressing issue of incremental few-shot object detection through weakly supervised class data augmentation. Utilizing only image-level labels from a large-scale classification dataset, our approach significantly enhances the generalizability of the hypernetwork, particularly towards novel categories. To counteract overfitting and preserve performance on pre-trained base categories, we freeze the backbone of the hypernetwork during training. Our method showcases superior performance, effectively balancing the trade-off between detecting novel and base categories. Future work may explore further optimization techniques and applications of this framework in real-world settings.

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

## A. Appendix

### A.1. Training & Evaluation Details

#### A.1.1. Training Details

The training process is composed of two parts, pre-training, and meta-training. During the pre-train, the hyper-parameters are set following FCOS [11]. SGD optimizer is adopted with a learning rate of 0.02 and the batch size is set to 32. The feature extractor along with heads are pre-trained for 45 000 iterations in total. Note that only the base classes are used for pre-train. During meta-train, 48 tasks are sampled at each iteration. The task length $t$ is set as 3. The learning rate is set to 0.005. The hypernetwork is optimized for 25 000 iterations in total.

#### A.1.2. Evaluation Protocol

Concretely, we follow the convention to use Average Precision (AP) to evaluate the results on COCO [28] dataset. Mean Average Precision (mAP) is used to evaluate the results on PASCAL VOC datasets [29]. By default, the results of base classes (using the classification weights from pre-training) and novel classes (using the classification weight predicted by the hypernetwork) are obtained by evaluating them separately. All ablation studies are conducted on COCO dataset and results are reported under different choices of shots.

| Grad-CAM Threshold | 0.3 | 0.4 | 0.5 | 0.6 | 0.7 | 0.8 | 0.9 |
|---|---|---|---|---|---|---|---|
| Novel Classes AP | 1.6 | 1.6 | 1.7 | **1.9** | 1.8 | 1.5 | 1.4 |

Table 4: **The study on the Grad-CAM threshold.** The novel classes AP is not sensitive to the threshold from 0.5 to 0.7.

| Class Aug. | MCF | Novel Classes AP | Base Classes AP |
|---|---|---|---|
| × | N/A | 1.1 | 38.4 |
| ✓ | × | 1.9 | 38.4 |
| ✓ | ✓ | **2.9** | 38.4 |

Table 5: **The effect of mis-classification filtering (MCF) on the final box AP**. MCF can improve the novel classes AP by 1%.

## A.2. Ablation Studies

To better supplement the results in our paper, we conduct additional experiments in this section. We use MS COCO [28] as the benchmark dataset.

In this section, we conduct several ablative studies on the WS-iFSD. By default, the class augmentation dataset used in the studies is constructed by 47 object categories randomly selected from ImageNet that do not overlap with COCO novel categories. We use 100 images per augmented category, and MCF is turned off. We report results on COCO.

### A.2.1. Is WS-iFSD Sensitive to Grad-Cam threshold?

Grad-CAM [1] outputs the highlighted heatmap in the input image, with a higher activation value on the relevant part for the target class. In order to get a proper pseudo bounding box of the object, we need to delicately choose a threshold value to crop the object out with the heatmap. Therefore, we perform an experiment to find the most suitable threshold value. We use Grad-CAM to generate pseudo bounding boxes for these images, which are used along with COCO dataset during the meta-training stage. The threshold for the bounding box ranges from 0.3 to 0.9. As shown in Table. 4, the threshold 0.6 appears to be the roughly best threshold value in our situation. However, it is worth noting that our method is not sensitive to the Grad-CAM threshold from 0.3 to 0.7.

### A.2.2. Does Mis-Classification Filtering Help?

Here, we verify the effectiveness of the proposed mis-classification filtering (**MCF**) technique, which removes the misclassified images by the Grad-CAM model from the augmented data. Qualitatively, Grad-CAM might fail to highlight the target object in misclassified images as shown in Figure. 3. To confirm the toxicity of these misclassified images in iFSD task, a quantitative study is conducted, and the results are presented in Table. 5. Line 1 (L1) is the baseline Base-iFSD. L2 and L3 denote WS-iFSD with and without MCF applied, respectively. Even without MCF, L2 still achieves a strong result over Base-iFSD. After applying MCF, the novel classes AP is further improved by another 1 AP, showing its effectiveness. The result also confirms that the misclassified images in class augmentation are harmful.

### A.2.3. The Impact of Novel Overlapping Classes

We study on the following 3 questions: (**1**) Would the existence of novel overlapping classes in augmented data be more beneficial to iFSD task? (**2**) In the case that novel overlapping classes exist in the augmented data, would adding more novel non-overlapping classes help? (**3**) If novel overlapping classes are not available, does simply increasing the number of novel non-overlapping classes help?

We conduct experiments and present results in Table 6. The first row in the table denotes the experiments using the augmented dataset containing the same 47 novel overlapping classes. The ex-

| # Aug. Classes | 0 | 47 | 100 | 200 | 500 |
|---|---|---|---|---|---|
| nAP w/ Novel Overlapping Classes | 1.1 | 2.2 | 2.3 | 2.3 | **2.5** |
| nAP w/o Novel Overlapping Classes | 1.1 | 1.9 | 2.2 | 2.2 | 2.4 |

Table 6: **The study on the number of classes and the effect of novel overlapping classes.** More classes are beneficial to iFSD task. Compared to non-overlapping classes, novel overlapping classes is more helpful. "nAP" denotes novel classes AP.

| # Images per Class | 0 | 50 | 100 | 200 | 500 | 1000 |
|---|---|---|---|---|---|---|
| Novel Classes AP | 1.1 | 1.8 | 1.9 | 2.0 | **2.1** | **2.1** |

Table 7: **The study on the number of images per augmented class.** Better performance are achieved with more images per augmented class, up to 500 images per class.

periments in the second row use the augmented dataset without any novel overlapping classes. In general, we could observe that results in the first row is better than the second row, which means including novel overlapping classes in augmented data would be more beneficial. We could also see that the novel classes AP increase gradually with the adding of non-overlap classes in both rows, which answers the rest two questions. Adding more novel non-overlapping classes to either novel overlapping classes or novel non-overlapping classes improves iFSD.

### A.2.4. The Effect of Augmented Class Frequency

We further investigate the effect of the number of images per augmented class by designing the following experiment. We gradually increase the number of images per augmented class from 0 to 1000. The results are shown in Table 7, showing a positive correlation between the novel class AP and the number of images per class. It is also worth noting that the growth of novel classes AP becomes saturated when the per class images number exceeds 500, indicating that adding more images might be unnecessary.

| Pre-train | | Meta-train | Novel | Base |
|---|---|---|---|---|
| Class Aug. | Reg. Loss on Aug. | Class Aug. | Class AP | Class AP |
| × | N/A | × | 1.1 | **38.4** |
| ✓ | ✓ | × | 1.5 | 37.2 |
| × | N/A | ✓ | 1.9 | **38.4** |
| ✓ | ✓ | ✓ | 1.9 | 37.2 |
| ✓ | × | × | 2.0 | 37.4 |
| ✓ | × | ✓ | **2.5** | 37.4 |

Table 8: **Augmentation Data Using stage.** Applying the class augmentation in pre-training stage could also benefit the novel classes AP, while the base classes AP would drop.

### A.2.5. The Impact of Data Augmentation on Pre-training

Previously, we confirmed class data augmentation improves the meta-training. In this study, we assess the impact of using it during the detector pre-training. The results are shown in Table 8. In the first block, we can observe adding the augmented data into either the pre-training (L2) or meta-training stage (L3) could help improve the novel class AP. However, enabling class augmentation on both stages fails to bring further improvement. Moreover, enabling class augmentation in pre-training leads to performance degradation on the base class AP.

We hypothesize that this is due to the low-quality bounding box annotations in the pseudo-labeled data, which is harmful and leads to performance degradation in the regression head. Therefore, we propose to disable the regression loss on the pseudo-labeled augmented data during the pre-training stage. The results are shown in the last two rows (L5, L6). We could observe that such a

| Framework | Pre-training Early Stopping | Hypernetwork Weight-sharing | nAP | bAP | AP |
|---|---|---|---|---|---|
| ONCE [26] | ✓ | × | 1.2 | 17.9 | 13.7 |
| Base-iFSD | ✓ | × | 1.0 | 24.2 | 18.4 |
| | ✓ | ✓ | **1.6** | 24.2 | 18.6 |
| | × | × | 0.8 | **38.4** | 29.0 |
| | × | ✓ | 1.2 | **38.4** | **29.1** |

Table 9: **Effects of pre-training early stopping and hypernetwork backbone weight-sharing.** Here "nAP" and "bAP" denotes novel and base classes APs, respectively. "AP" represents the AP across all classes. The last row is the final setting of our Base-iFSD, achieving the best AP with better base-novel AP trade-off.

| | Class Augmentation | | Novel Classes |
|---|---|---|---|
| Enable | Pseudo Label | Ground-truth Label | AP |
| × | N/A | N/A | 1.2 |
| ✓ | ✓ | | 4.0 |
| ✓ | | ✓ | **6.2** |

Table 10: **Class augmentation with ground truth ImageNet object detection annotation.** For class augmentation, we use all classes from ImageNet to construct the augmented data, where 500 images are randomly picked for each class.

simple technique could further improve the novel class AP and reduce the performance degradation on base classes.

### A.2.6. Ablation about Pre-training Early stopping and Hypernetwork Weight-sharing

In ONCE [26], it applies early stopping on detector pre-training, aiming at improving the generalizability of the model to novel classes. Also, it adopts a heavy-designed hypernetwork, where a dedicated ResNet-50 is used as the hypernetwork backbone. However, we argue that these designs are unnecessary and may bring critical issues instead. First, applying early stopping during pre-training scarifies the detector performance on base classes a lot. Second, using a dedicated overparameterized hypernetwork deteriorates its generalizability to novel classes.

We conduct experiments to verify our arguments and present results in Table. 9. In line 2 (L2) of the table, we design a Base-iFSD variant, which adopts pre-training early stopping and uses a dedicated FCOS feature extractor as the hypernetwork backbone. To verify the effectiveness of our backbone weight-shared hypernetwork, we compare L2 with L3. A significant performance improvement of 0.6 could be observed after applying hypernetwork weight-sharing. The same phenomenon could also be spotted comparing L3 and L4. To find the effect of pre-training early stopping, we can compare L2, L3 to L4, L5. Indeed, early stopping could offer a minor improvement on novel classes AP ($+0.3 \sim 0.4$). However, it deteriorates the base classes AP a lot (-14.2), which is not a good trade-off. Therefore, the setting of L5 becomes the final setting of our Base-iFSD, which achieves the best all classes AP and has a better novel-base AP trade-off.

### A.2.7. The Upper Bound of Using ImageNet Data as Augmentation

To explore the upper bound of using additional classes from ImageNet [27] in iFSD task, we conduct experiments that use the ground truth object detection annotation of ImageNet directly. Compared with our pseudo-labeled annotation, it is more accurate and may lead to better results. The results are presented in Table. 10. We could see that there is still a performance gap of 2.2% on novel classes AP between using ground-truth labels and the proposed weakly-supervised pseudo labels, indicating there is still space to improve.

## A.3. More Details of Our Approach

---

**Algorithm 1:** Meta-train the class weight hyper network in WS-iFSD.

---

**Notation:** $p(\mathcal{T})$: distribution over tasks on object detection and augmented pseudo-labeled datasets.
**Result:** A class weight hyper network that can generalize well to novel classes.
**Inner loop:** Produce the class weight given the support set.
**Outer loop:** Updates the class weight hyper network.
**while** *not converged* **do**
    Sample batch of tasks $\mathcal{T}_i \sim p(\mathcal{T})$;
    **for** $\mathcal{T}_i$ **do**
        Sample $K$ datapoints $\mathcal{D} = \{\mathbf{x}^{(j)}, \mathbf{y}^{(j)}\}$ from $\mathcal{T}_i$;
        Generate the class-specific weight $\omega_i$ through the weight hyper network $H$ with the
          input support set $\mathbf{x}^{(j)}$;
        Sample datapoints $\mathcal{D}' = \{\mathbf{x}^{(j)}, \mathbf{y}^{(j)}\}$ from $\mathcal{T}_i$ for the meta-update
    **end**
    Update parameters of the class weight hyper network:
    $\alpha_H \leftarrow \alpha_H - \beta \nabla_H \sum_{\mathcal{T}_i \sim p(\mathcal{T})} [\mathcal{L}_{cls}(X, Y; \alpha_H) + \mathcal{L}_{reg}(X, Y; \alpha_H) + \mathcal{L}_{center}(X, Y; \alpha_H)]$.
**end**

---

### A.3.1. Details About Pre-training

Following the original FCOS [11], the model is optimized by 3 loss terms, namely classification loss $\mathcal{L}_{cls}$, regression loss $\mathcal{L}_{reg}$ and centerness loss $\mathcal{L}_{center}$. The total training loss at the pre-training stage could be formulated as:

$$\mathcal{L}_{train} = \mathcal{L}_{cls}(X, Y; \alpha_F, \alpha_C) + \mathcal{L}_{reg}(X, Y; \alpha_F, \alpha_R) + \mathcal{L}_{center}(X, Y; \alpha_F, \alpha_T). \tag{2}$$

Here $X, Y$ denote the input images and the corresponding labels, respectively. $\alpha$ denotes the parameter. $F$, $C$, $R$, and $T$ represent the feature extractor, classification head, regression head, centerness head in our paper, respectively.

### A.3.2. Details about Meta-training

To supplement the illustration of our meta-training process in the paper, we further provide a detailed pseudo algorithm in Algo. 1. The meta-training consists of two loops. In the inner loop, a task $\mathcal{T}_i$ is sampled from $p(\mathcal{T})$. The hypernetwork $H$ predicts the class weight with $K$ support set samples as input, then tested on new samples from $\mathcal{T}_i$ and feedback at the outer loop. The hypernetwork is then updated by considering the detection loss on the new samples from the sampled task. Note that the weakly-labeled object annotations are mixed with the human-labeled annotations during meta-training.

## A.4. Future Work & Limitations

We notice that there's still improvement space in our work. Specifically, failure cases of pseudo labels generated by Grad-CAM could still be spotted even with our MCF applied, as shown in Figure. 4. To improve the annotation quality, a naive straightforward way might be considering replacing Grad-CAM [1] with a more advanced "visual explanation" model, such as Grad-CAM++ [83], XGrad-CAM [84], Ablation-CAM [85], Eigen-CAM [86].

Another future direction might be considering augmenting with a larger-scale classification dataset, like ImageNet-21K, which has more classes and might be more beneficial.

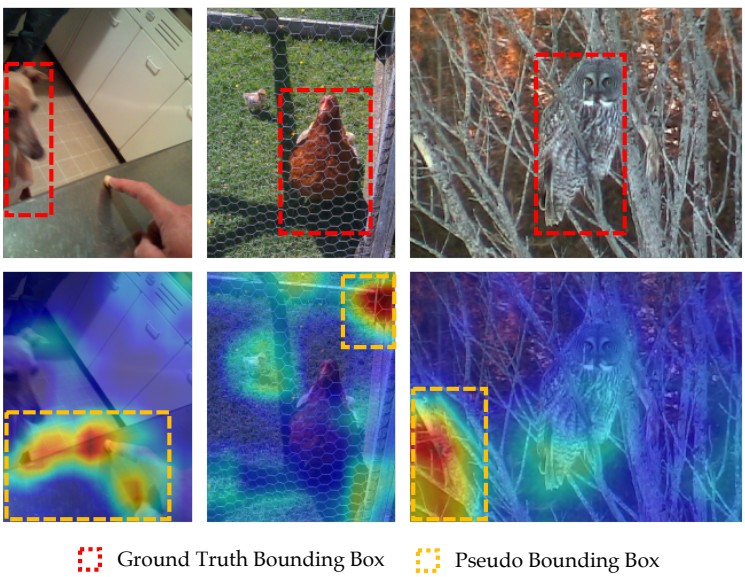

Ground Truth Bounding Box     Pseudo Bounding Box

Figure 4: **Failure cases of pseudo labels generated by Grad-CAM [1] with MCF.**

