# OpenReview forum: "WS-iFSD: Weakly Supervised Incremental Few-shot Object Detection Without Forgetting"
_CPAL.cc/2024/Conference — CPAL 2024 (Proceedings Track) Oral_

### Official Review · Reviewer_69Nq · 2023-10-06
**Review of Submission 36**

**Rating:** 7
**Confidence:** 3

**Review:**

**Overall Evaluation**
The primary innovation of this paper lies in bridging the gap between traditional object detection algorithms, which often lack adaptability to novel categories and introducing a weakly supervised class augmentation leveraging image-level labels. This approach underscores the paper's originality, and the empirical results demonstrate its efficacy.

*Pros*
1. The bottleneck of iFSD is effectively addressed through weak supervision, and the use of coarse-grained supervision by the authors is reasonable.
2. The WS-iFSD pipeline is thoughtfully designed, aligning seamlessly with the motivation. Additionally, the potential uncertainty in pseudo-annotation is well mitigated by MCF.
3. Overall, the paper is well-organized and easy to follow.

*Cons*
1. The analysis of computational complexity is missing. Considering that employing Pseudo Bounding Boxes from ImageNet through Grad-CAM facilitates better class augmentation, it might also increase time complexity. The authors should delve into this aspect, comparing the added complexity against performance enhancements.

2. The details for misclassification filtering are not comprehensive. One arising concern is whether the comparison of predicted labels against ground-truth labels was performed manually, which could be resource-intensive.

3. Given that class augmentation is derived from ImageNet, the extent of class overlap could be pivotal. While the authors touch upon the impact of novel overlapping classes, merely adding more classes, whether overlapping or not, for the novel classes AP seems somewhat straightforward and naive. A more thorough examination of the balance between the introduction of new classes and its effect on the base classes AP would be persuasive.

*Minor comments*
(1) Line 5: hypernetworkis -> hypernetwork is
(2) Line 156: hypernetworkand -> hypernetwork and

---

### Official Review · Reviewer_fvmq · 2023-10-10
**Review of the Manuscript on WS-iFSD**

**Rating:** 5
**Confidence:** 3

**Review:**

Strengths:

1. Innovation in Weakly Supervised Approach:
WS-iFSD integrates weakly localized objects using Grad-CAM, magnifying class and image data, offering a fresh, data-efficient approach to iFSD.

2. Performance Preservation Strategy:
Freezing the backbone and detection head during meta-training to preserve base-category detection performance is strategically sound.

3. Strong Empirical Evaluation:
Substantial improvements on benchmarks (MS COCO, PASCAL VOC) validate the method’s efficacy over current approaches like ONCE.

Weaknesses:

1.  Novelty of this paper is questionable. The entire framework seems a combination of existing techniques, e.g., Grad-CAM for weak object localization, backbnoe freezing, etc. Despite the effectiveness, the contribution of this paper is incremental.

2. Weak Localization Concerns:
The reliance on coarsely inferred bounding boxes could be a source of training noise and might impact detection reliability in complex environments. More analyses on the impact form those noise is beneficial.

3. Computational Complexity:
Absence of a detailed discussion regarding computational costs with increased class and image data might overlook practical deployment challenges.

3. Generalization Discussion:
Further exploration of the model’s generalization and performance across diverse detection scenarios and datasets is needed to establish robustness.

Overall:
The manuscript presents an interesting, data-augmented weakly supervised approach for iFSD, demonstrating marked improvements over established benchmarks. However the novelty and contribution of this work is questionable, and delving deeper into potential challenges with weak localization and computational complexities, along with a broader analysis of applicability, would fortify the research. The work is promising but warrants further exploration in specified areas for comprehensive insights and applicability.

---

### Official Review · Reviewer_fGfk · 2023-10-13
**Review comments**

**Rating:** 4
**Confidence:** 2

**Review:**

This paper proposes a framework for incremental few-shot object detection, which leverages a meta-learning approach with backbone and detection heads frozen, to generate class-specific codes, and it also introduces a weakly supervised approach for class augmentation technique with minimal requirement on image-level localization labels.

The novelty of this paper seems limited as the used meta-learning method, the freezing scheme and the class augmentation technique share much similarities with existing ones. The abstract states that "it outperforms the state-of-the-art ONCE approach 13 on the MS COCO dataset". However, ONCES [26] is published on CVPR 2020, which may not be viewed as a SOTA method.

---

### Official Review · Reviewer_bxCd · 2023-10-16
**Review for Submission36 WS-iFSD: Weakly Supervised Incremental Few-shot Object Detection Without Forgetting**

**Rating:** 7
**Confidence:** 3

**Review:**

Summary: This paper propose a new method for weakly spupervised incremental few-shot detection. They use a metra-learning approach and introduce a weakly supervised class augmentation. They outperforms traditional methods by a clear margin on MS COCO and PASCAL VOC.

Strength:
1. They have achived very good performance improvements in terms of both the number of new classess, and mAP.
2. It's interesting to study the problem that using images of a new category as the support set for meta-learning. In other words, I like the idea that applying meta-learning framework to incremental few-shot learning.
3. I like the idea that using a pre-trained classification model with Grad-CAM for data augmentation.


Weakenss:
1. In this paper, authors mainly compare their methods with ONCE,Feature-Reweight and MAML, what were proposed before 2020. I adverse authors to compare their method with some more advanced method.
2. Authors propose a new data augmentation method, and the Mis-Classification Filtering for remove the bad augmentations. However, no experimental results are probided to validate their effectiveness. Ablation on this is necessary.
3. In this paper, authors use models pretrained on ImageNet to introduce the knowledge from the new categories. Another popular method is to introduce new knowledge by using large-scale pretrained multi-modal models such as CLIP. To the best of my knowledge, the second  road has also achieved good performance. Does your method work better than such kind of method? Please discuss on it.

---

### Official Review · Reviewer_oZQJ · 2023-10-16
**WS-iFSD: Weakly Supervised Incremental Few-shot Object Detection Without Forgetting**

**Rating:** 6
**Confidence:** 3

**Review:**

The paper introduces an incremental few-shot object detection framework based on meta-learning networks. Specifically, the authors leverage the recent ONCE meta-learning network to learn class codes for new categories, enabling online addition of new classes for object detection without retraining. The paper highlights that training the meta-learning network alongside the detection network led to conflicts, resulting in limited generalization to new classes. To address this, the authors propose two strategies to enhance training effectiveness:

1. Utilizing Grad-CAM for coarse object localization in ImageNet,  significantly expanding the number of object categories and images during training to improve the meta-network's performance.
2. Freezing the detection network during the meta-training phase to retain the ability to detect base classes.

**Strengths**:

1. **Easy to follow**: The paper introduces an innovative approach to incremental few-shot object detection, leveraging meta-learning networks and class codes for new categories. This approach can potentially have significant practical implications.
2. **Clear Presentation**: The paper is well-structured and clearly presents the proposed framework, making it accessible to readers.

**Weaknesses**:

1. **Overlap in Training Data**: The potential overlap between ImageNet and COCO categories raises concerns about the effectiveness of the model in recognizing new classes. This issue needs to be addressed and clarified.
2. **Over-reliance on Class Codes**: The heavy dependence on class codes generated by the meta-network for detection poses potential challenges when the detection network struggles to extract features from new classes. The authors should explore alternative methods to mitigate this risk.

---

### Meta-Review · Area_Chair_QT55 · 2023-11-13

**Recommendation:** Accept (Poster)
**Confidence:** 4

**Metareview:**

The authors have properly addressed the concerns raised by reviewers and validated the advantages of combining existing techniques for few-shot object detection. Specifically, in the revision they have reported the computational complexity of the method.

---

### Decision · Program_Chairs · 2023-11-20

**Decision:**

Accept (Oral)

**Comment:**

The paper proposes a weakly supervised approach for incremental few-shot object detection. The paper contributes a novel class augmentation technique using Grad-CAM, and a performance preservation strategy by freezing the backbone and detection head during meta-training. The paper is well-written and the experiments are convincing. The reviewers raised some concerns about the novelty of the paper and the potential challenges of weak localization. The authors addressed these concerns in their rebuttal. The paper is ready to be accepted. In the camera ready, the authors should consider: (1) Providing a more detailed comparison with state-of-the-art methods; (2) Conducting more experiments to evaluate the robustness of the proposed approach to weak localization.

The action PC chair for this paper is Gintare Karolina Dziugaite, who made the decision after carefully reading the paper as well as the comments by all reviewers and AC. The decision is agreed by all PC chairs.